# Bioactive Compounds in Osteoarthritis: Molecular Mechanisms and Therapeutic Roles

**DOI:** 10.3390/ijms252111656

**Published:** 2024-10-30

**Authors:** Ahmed Maouche, Karim Boumediene, Catherine Baugé

**Affiliations:** UR7451 BIOCONNECT, Université de Caen Normandie, 14032 Caen, France; maouche.ah@hotmail.com (A.M.); karim.boumediene@unicaen.fr (K.B.)

**Keywords:** osteoarthritis, bioactive compounds, plant extracts, joint pain, inflammation, curcumin, *Boswellia serrata*, *Zingiber officinale*, avocado/soybean unsaponifiables, tea extracts, *Harpagophytum procumbens*, resveratrol

## Abstract

Osteoarthritis (OA) is the most common and debilitating form of arthritis. Current therapies focus on pain relief and efforts to slow disease progression through a combination of drug and non-drug treatments. Bioactive compounds derived from plants show significant promise due to their anti-inflammatory, antioxidant, and tissue-protective properties. These natural compounds can help regulate the inflammatory processes and metabolic pathways involved in OA, thereby alleviating symptoms and potentially slowing disease progression. Investigating the efficacy of these natural agents in treating osteoarthritis addresses a growing demand for natural health solutions and creates new opportunities for managing this increasingly prevalent age-related condition. The aim of this review is to provide an overview of the use of some bioactive compounds from plants in modulating the progression of osteoarthritis and alleviating associated pain.

## 1. Introduction

Osteoarthritis (OA) is the most common form of arthritis, affecting millions of people worldwide and a major cause of disability in the elderly. As a chronic degenerative disease primarily affecting the joints, its importance as an age-related condition is significant, but many other risk factors exist, such as metabolic syndrome, trauma/injury, sex, or genetic factors. Long regarded as a “wear and tear” disease characterized by loss of cartilage, OA is now recognized as a much more complex disease involving progressive degradation of cartilage, alteration of subchondral bone, and inflammation of synovial tissue. The major symptoms are pain and impaired movement leading to an increased sedentary lifestyle for OA patients, responsible for higher risk of cardiovascular diseases and mortality [1].

Although not classified as an inflammatory disease like rheumatoid arthritis, inflammation is nevertheless recognized as a key factor in the development of osteoarthritis. Inflammatory processes localized in the synovial membrane, known as synovitis, have been shown to trigger chronic inflammatory responses that exacerbate symptoms and accelerate joint damage. These responses include the release of pro-inflammatory cytokines such as Interleukin-1 beta (IL-1β) and Tumor necrosis factor-alpha (TNF-α) and the stimulation of pathways such as nuclear factor-kappa B (NF-κB) [2,3]. Subchondral bone is also a source of inflammatory mediators involved in the painful process of osteoarthritis and the degradation of the deep cartilage layer [4]. In addition to local inflammation, OA is associated with low-grade systemic inflammation. Interestingly, it has been observed that patients with increased expression of IL-1β gene in peripheral blood leukocytes had higher pain scores and decreased function, as well as a higher risk of radiographic progression of OA [5]. Furthermore, this low-grade systemic inflammation may promote comorbidity such as myocardial infarction, Alzheimer’s disease, or stroke [4].

There is currently no effective treatment for OA. Therapy consists of pain relief and attempt to slow disease progression [6]. The clinical treatment of osteoarthritis is represented by injection of drugs into the joint, such as hyaluronic acid or dexamethoasone, oral administration, such as non-steroidal anti-inflammatory drugs [7]. However, taking drugs to relieve osteoarthritis over a long period of time can harm various organs, in particular the kidneys, gastrointestinal tract, and cardiovascular system [8]. Consequently, there is an urgent need to develop a treatment strategy capable of relieving OA symptoms over a long period with minimal adverse effects in clinical practice.

Bioactive compounds from natural sources have promising potential due to their anti-inflammatory, antioxidant, and tissue protective properties [7]. These natural compounds could regulate the inflammatory processes and metabolic pathways involved in OA, thereby reducing symptoms and potentially slowing disease progression [9]. Therefore, investigating the efficacy of these natural agents in the treatment of osteoarthritis not only meets a growing demand for natural health solutions, but also creates new opportunities for the management of an increasingly prevalent age-related disease.

The aim of this review is to provide an overview about the use of natural bioactive compounds extracted from plants to modulate osteoarthritis progression and pain.

## 2. Physiopathology of Osteoarthritis

### 2.1. Cellular Mechanisms in Osteoarthritis

Osteoarthritis is associated to a dysregulation of all cells present in the joint, namely chondrocytes in the cartilage, synoviocytes in the synovial membrane, osteoclasts and osteoblasts in the bone. In a healthy joint, these cell populations work together to maintain joint homeostasis and function [10]. However, in OA joint, these cells are activated, leading to deregulation of homeostasis and inflammation.

Chondrocytes are the only cells found in healthy cartilage, and are responsible for maintaining the extracellular matrix by producing collagen (notably type II collagen) and proteoglycans (e.g., aggrecan), giving cartilage its tensile strength and elasticity [11]. During the early stages of OA, chondrocytes become activated, proliferate and undergo terminal differentiation. This is considered as an attempt to repair cartilage. However, the neosynthetised matrix is of poor quality with increased expression of type I collagen at the expense of type II collagen. In addition, inflammatory mediators present in OA joints promote the production of degradation enzymes (particularly the matrix metalloproteinases, MMPs), leading to cartilage destruction, as well as the local production of inflammatory mediators, contributing to increase inflammation in the joint [12]. Finally, in advanced osteoarthritis, chondrocytes die by apoptosis and cartilage matrix is no longer produced, leading to the inexorable disappearance of the tissue [13].

Synoviocytes, the cells that line the synovial membrane, are divided into two main types: fibroblast-like synoviocytes (FLS) and macrophage-like synoviocytes (MLS) [14]. FLS are involved in the production of synovial fluid and extracellular matrix components, while MLS play a role in immune surveillance and phagocytosis [14,15]. During OA, synoviocytes are the main producers of inflammatory mediators. The inflamed synovium produces a plethora of pro-inflammatory cytokines (e.g., IL-1β, TNF-α, IL-6) and chemokines, which perpetuate inflammation and attract more immune cells into the joint [16,17,18]. Inflammation leads to the production of excess synovial fluid, causing joint swelling and increased intra-articular pressure, which can exacerbate pain and stiffness [16]. Inflammation also promotes synovitis, characterized by hypertrophy and effusion of the synovial membrane, which further aggravates joint swelling and discomfort [19,20].

In addition, proinflammatory cytokines stimulate the production of MMPs by chondrocytes, leading to degradation of cartilage extracellular matrix. Inflammatory mediators such as prostaglandins and leukotrienes are also produced in larger quantity, increasing the sensitivity of nociceptors, and causing an increase in pain perception [20,21]. Synovitis, or inflammation of the synovial membrane, is therefore a key feature of OA pathology and plays an important role in disease progression. Synovial hyperplasia occurs when the synovial membrane thickens due to proliferation of synovial fibroblasts and infiltration by immune cells such as macrophages and lymphocytes. Around the synovial membrane, in the periarticular fat pads, adipocytes produce adipokines, which are signaling molecules involved in metabolic regulation and inflammation. These adipokines, such as leptin, adiponectin, and resistin, also contribute to the regulatory processes that maintain joint health [22].

Bone cells also contribute to osteoarthritis. Indeed, articular cartilage provides a smooth surface for movement, whereas subchondral bone provides stability and support. Dysregulation of bone homeostasis led thus inadequate function and health of joints. Bone is in perpetual renewal, thanks to a balance between bone resorption by osteoclasts and bone formation by osteoblasts [23]. During OA, several signals, including inflammation, activate osteoclasts leading to the formation of geodes that weaken subchondral bone [24]. Subsequently, osteoblasts also become more active, leading to tissue sclerosis. This bone remodeling contributes to the progression of osteoarthritis, but also to pain [25].

### 2.2. Molecular Mechanisms in Osteoarthritis

The pathophysiological processes leading to osteoarthritis are complex and involve a cascade of biochemical events that contribute to joint degradation and symptoms such as pain and stiffness. Understanding these mechanisms is essential for developing effective treatments.

Inflammation is one of the central mechanisms involved in disease progression and pain [26]. In the osteoarthritic joint, pro-inflammatory cytokines such as IL-1β and TNF-α are at the center of the inflammatory response [21,27]. As mentioned above, these cytokines are produced by all cell types in the joint, including chondrocytes, fibroblast-like synoviocytes, and infiltrating immune cells such as macrophages [16], and they will be able to act on all these cells in an autocrine and paracrine manner. IL-1β is the most studied cytokines in the OA physiopathology and is conventionally used to “model” OA in vitro since its concentration is increased in OA joint compared normal joint. IL-1β binds to its receptor on the surface of chondrocytes and synovial cells, activating signaling pathways such as NFκB and Mitogen-activated protein kinase (MAPK) [27,28]. These pathways lead to the transcription of genes involved in inflammation and catabolism, including those encoding matrix metalloproteinases, such as MMP-1, which degrades type I, II, and III collagens crucial to cartilage structure, MMP-3 [29], which degrades proteoglycans, laminin, fibronectin, and collagen, facilitating overall cartilage degradation, and MMP-13 [30], which specifically targets type II collagen, the main collagen type in articular cartilage, making it particularly destructive in OA [31]. Persistent NFκB activation in chondrocytes and synovial cells perpetuates a chronic inflammatory state, exacerbating tissue damage [27,32]. MAPK pathways, meanwhile, are activated by inflammatory cytokines and mechanical stress, leading to the phosphorylation and activation of downstream transcription factors such as AP-1 [33]. This activation results in the upregulation of genes associated with inflammation, ECM degradation, and chondrocyte apoptosis. Furthermore, p38 MAPK pathway is particularly involved in the production of proinflammatory cytokines and MMPs [34]. The crosstalk between MAPK and NF-κB pathways amplifies the inflammatory response, contributing to the chronic nature of OA [35]. Targeting NF-κB and MAPK signaling has been proposed as a therapeutic strategy to mitigate inflammation and slow disease progression, with promising results in preclinical studies [36]. IL-1β also promotes the production of nitric oxide (NO) and prostaglandins, contributing to inflammation and pain [19]. Similarly, TNF-α activates NF-κB and MAPK pathways, contributing to increased expression of inflammatory mediators and cartilage-degrading enzymes [33]. TNF-α also increases the sensitivity of nociceptive neurons, contributing to pain perception in OA. A hallmark of the inflamed joint is the overexpression of cyclooxygenase-2 (COX-2), which leads to elevated production of prostaglandin E2 (PGE2) [37]. Also, the 5-lipoxygenase (LOX-5) pathway contributes to the synthesis of leukotrienes, pro-inflammatory mediators that further exacerbate joint inflammation in OA [38].

Oxidative stress, the result of an imbalance between the production of reactive oxygen species (ROS) and their clearance by antioxidant defense system, is elevated in OA cartilage and is a major cause of chronic inflammation [39]. ROS are oxygen containing free radicals including hydrogen peroxide (H_2_O_2_), hydroxy radical (OH^−^), superoxide anion (O_2_^−^) and nitric oxide (NO) and possess unpaired electrons, making them unstable and highly reactive [40]. They are normally produced in cells at low levels and are essential for maintaining cellular homeostasis and function. In the cartilage and chondrocytes of osteoarthritis patients, ROS levels are highly upregulated, and this overproduction of ROS and induction of oxidative stress is proving to be one of the main contributors to OA pathogenesis [41]. The imbalance in redox leads to increased expression of inflammatory cytokines and chemokines, which cause oxidation of cellular macromolecules such as proteins, lipids and DNA impairing their function [39]. The main sites of ROS production include mitochondria, peroxisomes and other membranous structures containing NADPH oxidases (NOX), Xanthine Oxidase (XO) and Nitric Oxide Synthase (NOS). In joint tissues, the expression of inducible NOS (iNOS), one of three NOS isoforms, is strongly upregulated in chondrocytes in response to the stimulation of inflammatory cytokines such as IL-1β or TNFα, promoting the production of NO which increases inflammation by activating NFκB pathway, leading in turn increased production of IL-1β and TNFα [39]. Furthermore, excessive amounts of ROS function as secondary messengers and promote cartilage degradation by inducing the expression of matrix degrading proteases, reducing extracellular matrix synthesis and inducing chondrocyte apoptosis [42]. On the contrary, the expression of antioxidant defense system proteins, including superoxide dismutases (SOD), catalase and glutathione peroxidase (Gpx), is downregulated in OA joints, highlighting the redox imbalance in osteoarthritic cartilage [43]. Likewise, the expression of Nuclear factor erythroid 2–related factor 2 (Nrf2), a master transcription factor regulator of the cellular antioxidant defense system, is dysregulated in OA and its suppression leads to increase disease development in a mouse model of OA induced by destabilization of medial meniscus (DMM) induced OA, confirming the importance of oxidative stress regulation in osteoarthritis [39].

Recent studies have shown that oxidative stress and the consequent generation of ROS, as well as inflammation signals (IL-1β, TNFα) are capable of promoting osteoclast differentiation [44], altering subchondral bone homeostasis, which contributes directly and indirectly to cartilage destruction and pain. The exact role of subchondral bone in the onset and progression of osteoarthritis remains unclear. However, there is growing evidence that subchondral bone lesions, including bone marrow edema and angiogenesis, develop earlier than cartilage degeneration [45]. Osteoclasts, the main cells governing bone resorption, play a crucial role in the homeostasis of subchondral bone through the secretion of degradation enzymes, immunomodulation, and cell signaling pathways [44]. Multiple signaling pathways and molecules are involved in the recruitment of these subchondral osteoclasts. For instance, RANKL binds to the RANK receptor on the surface of osteoclasts and activates them, a critical step in the migration of osteoclasts into bone tissue. Several chemokines and chemotactic proteins, such as CCL2 and CX3CL1, can be produced in subchondral bone and attract osteoclasts to these regions. Besides, hyperactive osteoclasts secrete proteases and degrading enzymes, such as MMPs, which break down the cartilage matrix and lead to the structural destruction of articular cartilage. They also affect angiogenesis and innervation of subchondral bone, accelerating articular cartilage damage and causing joint pain [44].

In addition to inflammation and oxidative stress, other mechanisms also play major role in OA. Recent studies have established the important role of Wnt signaling in OA pathogenesis, and have showed that the expression of members of the Wnt signaling pathway is upregulated in OA cartilage, synovium and subchondral bone [46]. Studies have revealed multiple functions of the Wnt signaling cascade in the maintenance of joint homeostasis, reflected in its ability to affect bone formation, endochondral ossification, bone growth and repair, and joint development. The Wnt cascade is tightly regulated, and its role in OA has yet to be fully elucidated. Thus, both overexpression of Wnt signaling and blockade of its activation may result in cartilage damage and bone erosion; making necessary to investigate this signaling pathway in more detail before developing specific therapeutic interventions targeting this pathway [47,48].

Likewise, transforming growth factors beta (TGF-β) and bone morphometric proteins (BMPs) perform essential functions during OA. TGF-β and BMPs belong to the same family, share structural similarities and transduce signals via SMAD-dependent and -independent pathways. However, they recruit different receptors to activate independent sets of SMAD proteins, thus establishing the molecular basis for their diverse functions. BMPs promote osteogenesis, osteoclastogenesis, and chondrogenesis at all stages of differentiation, while TGF-βs play different roles depending on the stage. Furthermore, BMPs and TGF-β have opposite functions in articular cartilage homeostasis. Hence, TGF-β signaling plays dual or opposite effects depending on context, making it complex to understand their respective roles in OA [49].

Thereby, osteoarthritis is a complex disease that affects all the joints through multiple pathways.

## 3. Natural Bioactive Compounds Against OA

In recent years, significant progress has been made in the treatment of osteoarthritis using medicinal plants. Numerous investigations have been carried out in vitro (Table 1) and in vivo (Table 2) and several clinical trials (Table 3) have revealed that certain medicinal plants or their active substances could be capable of relieving osteoarthritis.

### 3.1. Curcuma longa/Curcumin

*Curcuma longa*, commonly known as turmeric, is a flowering plant belonging to the Zingiberaceae (ginger) family and is native to Southeast Asia and Indian subcontinent [50]. The plants are cultivated and harvested each year for their rhizomes, which are consumed as a dried powder called curcuma, the spice which gives curry its yellow color [51]. It is widely used in culinary and medicinal traditions, notably in Ayurvedic medicine. The main components of turmeric roots are volatile oils and curcuminoids; the latter include diferuloylmethane (82%), demethoxycurcumin (15%), and bisdemethoxycurcumin (3%), three major components of commercial curcumin. Curcumin is an orange-yellow constituent of turmeric, with important anti-inflammatory and antioxidant properties [52]. Curcumin is a highly lipophilic molecule, with poor absorption, rapid metabolism, and low bioavailability limiting its biological activity [53]. It is generally taken orally as a nutritional supplement, but its low bioavailability represents a major challenge to its efficacy [54]. Nonetheless, numerous studies have demonstrated anti-OA activity.

#### 3.1.1. In Vitro

The effect of curcumin in chondrocytes has been widely documented. Mechanisms include regulation of oxidative stress, inhibition of chondrocyte apoptosis, autophagy, and inhibition of the NF-κB signaling pathway.

Curcumin acts primarily by inhibiting NF-κB activation, which significantly reduces the secretion of inflammatory cytokines such as TNF-α, IL-1β, and IL-6, as well as the expression of matrix metalloproteinases such as MMP-3, MMP-13, and MMP-1 in human chondrocytes [55,56]. This ability of curcumin to suppress MMP expression and secretion helps prevent extracellular matrix degradation [57]. In addition, curcumin enhances type II collagen and aggrecan biosynthesis in rat chondrocytes, conferring tensile strength and elasticity to the tissue [58]. This anabolic effect promotes the maintenance and repair of damaged OA cartilage. Additionally, curcumin counteracts the expression of pro-apoptotic molecules such as caspase-3 and genes related to antioxidant and cytoprotective pathways, particularly the Nrf2 pathway [55,56].

Besides its effects in chondrocytes, curcumin reduces the production of pro-inflammatory cytokines and PGE2 in human synoviocytes [59]. Additionally, curcumin’s antioxidant properties help reduce oxidative stress in synovial fluid, further mitigating inflammation and cell damage [60]. Through these mechanisms, curcumin can potentially reduce synovial hyperplasia and contribute to reduce pain and swelling in affected joints.

Furthermore, curcumin has been shown to affect function of bone cell function by modulating the RANKL-mediated signaling pathway, thereby inhibiting osteoclastogenesis in different mouse cell line (C3H and 7F2), which may contribute to reduce geode formation and pain [44,61,62,63]. In addition, curcumin enhances bone formation by stimulating the expression of bone morphogenetic proteins (such as BMP2) and SMAD signaling, which promote bone matrix formation and mineralization [64,65]. These actions help maintain bone density and integrity, and prevent subchondral bone loss in advanced osteoarthritis.

#### 3.1.2. In Vivo

Curcumin has shown promising therapeutic effects on osteoarthritis in various animal models. Studies suggest that curcumin significantly alleviates joint inflammation, reduces pain, and protects cartilage integrity in experimental OA models.

In animal models where OA was induced using monoiodoacetate (MIA), curcumin administration led to significant reductions in joint swelling and cartilage degradation [66]. Histological assessments revealed that curcumin-treated rats showed reduced inflammatory cell infiltration and improved chondrocytes organization compared to untreated osteoarthritic group. This was accompanied by a significant decrease. in pro-inflammatory cytokines such as IL-6, IL-1β, and TNF-α in synovial fluid, underlining curcumin’s strong anti-inflammatory action [67].

Micro-computed tomography (μCT) analysis reveals that curcumin treatment attenuates subchondral bone remodeling, reducing bone resorption and osteophyte formation [68]. This is in line with the effects observed in vitro, and indicates that curcumin not only protects cartilage, but also preserves the integrity of subchondral bone, which is crucial for preventing the progression of osteoarthritis.

#### 3.1.3. Clinical Trial

Several clinical trials have demonstrated curcumin’s anti-inflammatory and analgesic properties in osteoarthritis. For instance, Lopresti et al. (2022) conducted an 8-week, randomized, double-blind, placebo-controlled study involving 101 adults with knee OA [69]. Participants who received 500 mg of standardized curcumin extract twice daily showed a significant reduction in knee pain compared to the placebo group. Additionally, curcumin supplementation was associated with improved performance in physical function tests. Furthermore, 37% of participants in the curcumin group reduced their use of analgesics compared to only 13% in the placebo group [69]. In addition, Singhal’s study suggests that bioavailable turmeric extract is as effective as paracetamol in reducing pain and other symptoms of knee osteoarthritis, and is safe and more effective in reducing C-reactive Protein (CRP) and TNF-α [70].

Several systematic literature reviews and meta-analysis have evaluated the beneficial effects of curcumin on osteoarthritis. Most conclude that curcumin has beneficial effects on knee OA [71,72,73]. For instance, Hsiao et al., reviewing 11 randomized controlled trials with a total of 1258 participants, demonstrated that curcumins are associated with better pain relief than NSAIDs, suggested that low- and high-dose curcumins have similar effects on pain relief and adverse events in knee OA, and recommend to use curcumin as adjunctive treatment in knee OA [74].

### 3.2. Boswellia serrata/Boswellic Acid

*Boswellia serrata* gum resin (also called Indian Frankincense), an oleo-gum resin derived from the Boswellia serrata tree native to the Indian highlands, has historically been used as a remedy for chronic inflammatory diseases. The gum resin of *Boswellia serrata* contains monoterpenes, diterpenes, triterpenes, tetracyclic triterpene acids, and pentacyclic triterpene acids, called boswellic acids (BAs). There are six major boswellic acids, mainly keto-β-boswellic acid (KBA) and 3-O-acetyl-11-keto-β-boswellic acid (AKBA) which were described as responsible for the anti-inflammatory and antioxidant activities of the Boswellia gum resin. AKBA and KBA have in particular a potent 5-lipoxygenase inhibitory activity [75,76]. *Boswellia serrata* is frequently available in extract form, which is a concentrated and standardized version of the resin with elevated levels of boswellic acids, especially AKBA, to optimize its anti-inflammatory effects. These extracts are more readily absorbed by the body, making them especially effective for therapeutic use in supplement form.

#### 3.2.1. In Vitro

*Boswellia serrata* extracts (BSE) exhibits potent anti-inflammatory and chondroprotective effects in vitro targeting chondrocytes, synoviocytes, and bone cells. In rat chondrocytes, BSE are able to inhibit the production of matrix metalloproteinases, in particular MMP-3 and MMP-13, and enhance the expression of type II collagen and tissue inhibitors of metalloproteinases (especially TIMP-1 and TIMP-3), that may limit cartilage degradation and promote cartilage repair and regeneration [77]. Furthermore, BSE are able to reduce synovial inflammation by inhibiting NF-κB pathway, and modulate the activity of the COX-2 in mouse primary osteoblast cell culture and Mouse macrophage cell line (RAW 264.7) [78,79] Moreover, boswellic acids inhibit the 5-LOX pathway, thus reducing leukotriene production, and consequently lowering the overall inflammatory response in osteoarthritic cells [80]. In bone cells, BSE down-regulate RANKL, inhibiting osteoclastogenesis [79], and reduce the expression of vascular endothelial growth factor (VEGF), which is involved in pathological angiogenesis, that could help to reduce subchondral bone damage during osteoarthritis [81].

#### 3.2.2. In Vivo

Several animal models have demonstrated the beneficial effects of BSE in osteoarthritis. For example, Alluri et al. found that a standardized BSE containing at least 30% boswellic acid and not less than 5% KBA alleviated pain and protected cartilage in a rat model of MIA-induced osteoarthritis [82]. This finding was corroborated by Shin et al., who observed that oral administration of another BSE enriched with 30% AKBA significantly reduced inflammatory responses, decreased cartilage degradation, and lessened joint pain in a similar MIA-induced OA rat model [38]. At the molecular level, they demonstrated that BSE suppresses the expression of inflammatory enzymes, such as cyclooxygenase COX-2 and lipoxygenase LOX-5, leading to a significant reduction in PGE2 and leukotriene B4 levels. Additionally, BSE was shown to inhibit cartilage deterioration by downregulating MMPs [38].

Kim et al. also explored the effects of BSE (standardized for combined AKBA and KBA content at a concentration of 71 mg/g) on a rat model of MIA-induced osteoarthritis. Their findings confirmed the therapeutic potential of BSE, as diet supplementation in osteoarthritic rats resulted in reduced tissue injury, minimized cartilage destruction, and decreased MIA-induced roughness on the articular cartilage surface. BSE appeared to exert its anti-inflammatory effects by inactivating NFκB, blocking the CASP3 pathway associated with FADD and Bax to suppress chondrocyte apoptosis, and inhibiting the phosphorylation of the JNK pathway, a key signaling cascade involved in stress and inflammatory responses, to reduce MMP activation [77].

Recently, Choi et al. investigated the preventive effects of a standardized gum-resin extract of *Boswellia serrata* (BSRE) in a rat model of MIA-induced osteoarthritis. The BSRE, which was rich in AKBA and KBA (totaling 110 mg/g), was administered orally 14 days before the induction of osteoarthritis, with the effects of the preventive treatment assessed on day 21. The study demonstrated that BSRE alleviated knee joint swelling, reduced cortical bone erosion, and improved histomorphological changes in joint cartilage. Mechanistically, these anti-osteoarthritis effects were associated with upregulated expression of COL2A1 and aggrecan in the articular cartilage, along with a decrease in inflammatory mediators (NO, PGE2), cytokines (IL-6, IL-1, TNFα), and cartilage-degrading enzymes (MMP-3 and MMP-13) in serum. In the knee joint synovium, BSRE also suppressed the mRNA expression of inflammatory mediators (iNOS, COX-2, 5-LOX), cytokines (IL-1, IL-6, TNFα), and degrading enzymes (MMPs) [83].

Taken together, these findings highlight the potential of BSE as a preventive and therapeutic agent for alleviating osteoarthritis-induced knee joint problems, cartilage damage, and inflammation.

#### 3.2.3. Clinical Trial

A series of randomized, placebo-controlled clinical studies have suggested that various standardized preparations of *Boswellia serrata* gum resin extracts are effective and safe alternative interventions for managing OA pain [84,85,86,87,88,89]. The effectiveness and safety of Boswellia and its extracts for OA have been confirmed by several systematic reviews and meta-analyses [76,90,91].

In 2020, a meta-analysis based on seven trials involving 545 patients, conducted by Yu et al., showed that Boswellia and its extracts may be a novel treatment for patients with OA [76]. They reported that BSE was effective in reducing pain, improving joint function, and reducing inflammation in OA patients. They recommended that the duration and dosage of BSE treatment should be at least 100–250 mg for four weeks. However, their conclusions need to be nuanced, and they themselves highlighted the necessity of new studies to confirm or refine the results of their work [76]. More recently, another meta-analysis, performed on nine randomized clinical trials with 712 participants, also concluded that BSE supplementation is effective for managing OA symptoms [92].

Regarding the safety of *Boswellia serrata*, studies have shown that its extract does not have toxic side effects even at high doses [93,94,95], indicating that the active compound in Boswellia extract (AKBA) is safe based on current evidence [76]. BSE have been described to have an efficacy comparable to conventional treatments like non-steroidal anti-inflammatory drugs (NSAIDs) but with fewer gastrointestinal side effects, enhancing their clinical applicability for long-term use [96].

### 3.3. Zingiber officinale

Commonly known as ginger, *Zingiber officinale* is a member of the Zingiberaceae family used as a spice and herbal remedy for over 3000 years. Its rhizome is particularly rich in bioactive compounds, which have been utilized in Ayurvedic and Chinese medicine. The pharmacological properties of ginger are attributed to numerous active phytocompounds, primarily phenolics and terpenes.

The rhizomes of ginger contain two distinct types of compounds: non-volatile oleoresin, which gives ginger its characteristic pungency, and volatile essential oils. Oleoresin is the source of ginger’s key physiologically active substances, including gingerols, shogaols, paradols, and zingerone. The volatile essential oils primarily consist of sesquiterpenes, such as α-zingiberene, along with smaller amounts of monoterpenes, including borneol, 1,8-cineole, β-linalool, and geranial [97].

6-gingerol is the predominant compound in fresh, unprocessed rhizomes and is known for its anti-inflammatory and analgesic properties. However, 6-gingerol is unstable at higher temperatures, and dehydration processes convert it into 6-shogaol. Research indicates that 6-shogaol, which is the primary compound in dried ginger rhizomes, possesses also significant biological benefits [98]. The differing compositions of fresh and dried ginger impact the effectiveness of various formulations and should be carefully considered when selecting an active compound or ginger extract for medicinal use. These compositional differences are primarily influenced by factors such as storage conditions, thermal processing, and extraction methods [97].

#### 3.3.1. In Vitro

The anti-arthritic potential of *Zingiber officinale* Roscoe has been extensively documented in a recent review [97]. The authors of this manuscript conclude that many of ginger’s bioactive components may have therapeutic benefits in treating osteoarthritis by targeting chondrocytes and synoviocytes. For example, Ruangsuriya et al. found that zingerone reduced the levels of p38 and JNK in SW1353 cell cultures and suggest that it may help prevent cartilage destruction by suppressing MMP-13 expression [99]. Additionally, zingerone treatment downregulated genes involved in inflammation, including TNF-α, IL-6, and IL-8 [100]. Similarly, 6-shogaol inhibits NO production, decreases the expression of MCP-1 and IL-6, and prevents apoptosis in human chondrocytes [101]. In a human cell line of chondrocytes (C28I2), it has also been showed that ginger derivatives enhance the mRNA expression of genes associated with antioxidant enzymes, including superoxide dismutase (SOD1), glutathione peroxidase (GPX1, GPX3, GPX4), and catalase (CAT), and reduce markers of mitochondrial cell death, such as the Bax/Bcl-2 ratio and caspase 3 activation [102]. In addition, an aqueous extract of ginger decreased PGE2 and NO concentrations in healthy and osteoarthritic chondrocytes [103,104].

In addition to its effects in chondrocytes, ginger extracts also have anti-inflammatory effects in human synoviocytes. They can inhibit the expression of pro-inflammatory cytokines such as IL-1β, TNF-α, and COX-2 [105,106], as well as reduce the expression of the chemokines MCP-1 and IP-10 in synoviocytes. The combined extract of Alpinia galanga and *Zingiber officinale* was particularly effective, suggesting a synergistic enhancement of anti-inflammatory effects [107].

While most in vitro studies focus on the anti-inflammatory effects of *Zingiber officinale* extracts in chondrocytes and synoviocytes, there is emerging evidence suggesting that these extracts might also impact subchondral bone cells. The inhibition of inflammatory pathways and the reduction of pro-inflammatory cytokines by ginger extracts imply potential benefits for subchondral bone health. Additionally, one study has shown that ginger hexane extracts inhibited osteoclast differentiation in RAW264.7 cells [108,109].

#### 3.3.2. In Vivo

In vivo studies have highlighted the potential of *Zingiber officinale* in preserving joint health. In various animal models, such as rat and mice OA models induced with collagenase, ginger extracts significantly reduce cartilage degradation. These effects are achieved through the decreased activity of MMPs and lower levels of pro-inflammatory cytokines like IL-1β and TNF-α and PGE2. Doses of ginger extracts used in these studies typically range from 100 to 200 mg/kg body weight per day, administered orally [98,110].

Moreover, ginger has shown to effectively reduce synovial inflammation. There is decreased synovial hyperplasia and reduced inflammatory cell infiltration in the synovial membrane observed by histological analysis [98]. The suppression of pro-inflammatory cytokines and mediators in synoviocytes contributes significantly to alleviating joint inflammation and pain. Furthermore, ginger exerts protective effects on subchondral bone [111].

#### 3.3.3. Clinical Trial

Clinical trials have consistently demonstrated that ginger supplementation can significantly reduce pain and improve function in patients with osteoarthritis. Rondanelli et al. (2020) found that patients receiving a lecithin formulation of ginger showed substantial improvements in pain scores and physical function compared to a placebo [112]. A meta-analysis by Mathieu et al. (2022) confirmed the safety and efficacy of ginger for managing osteoarthritis symptoms. This analysis, which included three randomized controlled trials (166 patients supplemented with ginger compared to 164 placebo controls), highlighted that ginger not only reduces pain but also enhances quality of life without significant adverse effects [113]. Comparative studies, such as those by Afshar et al. (2022), indicated that ginger can be as effective as conventional NSAIDs in reducing pain and inflammation in osteoarthritis patients [114]. Furthermore, ginger has a more favorable safety profile, with fewer gastrointestinal side effects compared to NSAIDs [114]. In a clinical study by Mozaffari-Khosravi et al. (2016), a standardized dose of 500 mg of ginger extract was administered twice daily to participants over three months. The results showed a significant reduction in pain intensity and improvements in functional mobility, along with a notable decrease in serum levels of pro-inflammatory cytokines such as TNF-α and IL-1β [115]. Beside oral administration, topical application of ginger gel has also been tested and found to effectively diminish knee pain in patients with OA [116]. These findings highlight the potential of ginger as an effective adjunct therapy in managing osteoarthritis [97].

### 3.4. Avocado/Soybean Unsaponifiables (ASU)

Avocado-soybean unsaponifiable (ASU) is a natural plant extract made from avocado and soybean oils, consisting of the remainder of the saponified portion of the product that cannot be made into soap after saponification (approximatively 1%). The main components of ASU are phytosterols, beta-sitosterol, canola stanols, and soya stanols, which are rapidly incorporated into cells. ASU is a complex mixture of many compounds including fat-soluble vitamins, sterols, triterpene alcohols, and possibly furan fatty acids. ASU is formed of one-third avocado and two-thirds soybean unsaponifiables. The active components remain unknown [117]. ASU have extensively been documented for its anti-inflammatory, antioxidant, and analgesic properties, which collectively contribute to joint health preservation. It is a member of the symptomatic slow-acting drugs for OA, which are effective in relieving OA symptoms in patients. However, there are contradictory findings regarding whether ASU exerts structure-modifying effects [118].

#### 3.4.1. In Vitro

ASU possesses chondroprotective, anabolic, and anticatabolic properties. In chondrocytes, ASU can inhibit the production of pro-inflammatory cytokines (IL-1β, IL-6, IL-8), chemokines, PGE2, nitric oxide, and matrix metalloproteinases such as MMP-13, while promoting the synthesis of extracellular matrix components like collagen and aggrecan [117,119,120]. This dual action not only mitigates cartilage degradation but also fosters tissue repair and maintenance. Similarly, in human synoviocytes, ASU reduces the expression of inflammatory mediators and enzymes, thereby alleviating synovitis and protecting the synovial membrane [118]. Furthermore, ASU exerts protective effects on rat subchondral bone by inhibiting MMP-13 activity and nitric oxide synthase (iNOS), preventing bone resorption, and maintaining bone density [118]. These combined effects underscore ASU’s potential in slowing OA progression and enhancing joint function, making it a valuable addition to OA treatment strategies.

#### 3.4.2. In Vivo

ASU exhibits significant protective effects in various in vivo models of osteoarthritis. In an MIA-induced OA rat model, ASU demonstrated structure-modifying effects by improving synovial membrane pathology, reducing cartilage destruction, and mitigating subchondral bone changes through the reduction of oxidative stress and inflammation. Additionally, ASU suppressed catabolic factors that disrupt extracellular matrix homeostasis in OA cartilage [118]. Similar results have been observed in other OA animal models. For example, in an experimental canine OA model induced by anterior cruciate ligament sectioning, ASU treatment reduced the development of early OA lesions in the synovial membrane, cartilage, and subchondral bone [121]. In sheep, ASU treatment following cartilage injury enhanced articular regeneration, as evidenced by improved toluidine blue staining after six months compared to untreated animals [122]. These improvements were attributed to the downregulation of catabolic processes and the upregulation of anabolic activity in cartilage by ASU. Collectively, these findings provide clues of the therapeutic potential of ASU in managing OA by not only reducing pain and inflammation but also by protecting and preserving joint structures at multiple levels [118].

#### 3.4.3. Clinical Study

Several clinical studies have evaluated the beneficial effects of ASU in OA patients [123,124,125]. However, we can identify only two systematic reviews and meta-analysis of randomized placebo-controlled trials, which were published in 2008 and 2019 [126,127]. These meta-analyses suggested a beneficial effect of ASU treatment in symptomatic knee OA but not in hip OA. Thus, in the most recent meta-analysis, Simental-Mendia et al., conclude that ASU therapy significantly decreased both VAS and Lequesne index in knee OA but not in hip OA [127], confirming the conclusion of Christensen et al. which were based on four trials (all supported by the manufacturer) including 664 OA patients with either hip or knee, which received ASU (336 patients) or placebo (328 patients) [126]. Both analyses concluded in the safety of treatment: adverse events were similar in patients receiving ASU therapy or placebo [127].

### 3.5. Tea Extracts

Tea (*Camellia sinensis*), a traditional beverage, is believed to reduce the risk of various diseases due to its antioxidant and anti-inflammatory properties [128]. The effects of green tea, which is a rich source of polyphenols, mainly epigallocatechin 3-gallate (EGCG) which corresponds to 30–60% [129,130], have been extensively studied for their role in preventing joint disease. Fewer studies have investigated the properties of black tea compounds, primarily theaflavins (TFs), which include theaflavin-3-gallate, theaflavin-3′-gallate, and theaflavin-3-3′-digallate [131].

#### 3.5.1. In Vitro

Numerous in vitro studies have highlighted the beneficial effects of EGCG in chondrocytes. EGCG has been shown to inhibit the degradation of proteoglycans and type II collagen in both bovine and human cartilage explants [132]. This inhibition is achieved through the downregulation of matrix metalloproteinases [133]. Additionally, EGCG suppresses the production of inflammatory cytokines such as IL-1β [134] and TNF-α [135]. Furthermore, EGCG treatment reduces IL-1β-induced nitric oxide production by inhibiting inducible nitric oxide synthase (iNOS) expression in human chondrocytes [136]. Moreover, EGCG selectively inhibits the activation of MAPKs such as c-Jun N-terminal kinase (JNK), which play a crucial role in the inflammatory signaling pathways within chondrocytes [134]. In addition to its chondroprotective effects, EGCG alleviates synovial inflammation [137].

Other tea components also have the ability to protect chondrocytes. For example, theaflavins have been shown to reduce the expression of catabolic and proinflammatory factors, including matrix metalloproteinase-13 and interleukin-1, in chondrocytes [131]. Additionally, theaflavin-3,3′-digallate (TF3) significantly inhibits chondrocyte ferroptosis by activating the Nrf2/Gpx4 signaling pathway, indicating that TFs may serve as potential therapeutic supplements for osteoarthritis treatment [138].

#### 3.5.2. In Vivo

Green tea and its active compound, EGCG, have been shown to protect against cartilage degradation in various animal OA models [136]. EGCG helps attenuate articular cartilage degeneration by enhancing autophagy, reducing inflammation, and inhibiting matrix degradation. In post-traumatic osteoarthritis rat models, intra-articular injections of EGCG resulted in decreased cartilage destruction and improved histological scores [137]. Furthermore, green tea polyphenols have been found to positively influence subchondral bone remodeling. They mitigate abnormal subchondral bone changes by inhibiting osteoclast activity and promoting osteoblast differentiation, thereby maintaining bone integrity [139]. Studies involving dietary supplements containing green tea extracts in canine OA models have indicated reduced pain and improved joint function, as reported by both veterinarians and owners [140]. Additionally, intraperitoneal injections of EGCG significantly slows OA disease progression and relieve OA-associated pain as indicated by higher locomotor behavior in a posttraumatic OA mouse model [141]. Another study performing in a spontaneously occurring OA model in guinea pigs also showed that intra-articular injection of EGCG exerted an anti-OA effect by reducing ECM degradation, cartilage inflammation, and cell senescence [142]. Furthermore, EGCG has been shown to enhance subchondral bone quality by inhibiting osteoclastogenesis, the process of bone resorption mediated by osteoclasts, thus contributing to the overall structural integrity of the joint [136] Green tea polyphenols also exhibit chondroprotective effects by preventing chondrocyte apoptosis and senescence, primarily through the regulation of the Nrf2 pathway [143].

Regarding black tea, recent studies in a rat OA model induced by destabilized medial meniscus (DMM) have shown that theaflavin-3,3′-digallate (TFDG) can protect chondrocytes in vivo. OA rats receiving TFDG exhibited lower cartilage breakdown and expressed higher levels of COL2 and Nrf2 compared to those in the DMM group [144].

#### 3.5.3. Clinical Studies

Although evidence from cell, tissue explant, and animal studies strongly suggests that green tea and EGCG could mitigate cartilage degradation by targeting multiple aspects of joint health during OA progression, there are only limited studies testing their anti-arthritic effects in humans [136].

Several studies have reported an association between high green tea intake and a lower incidence of OA [145,146]. Additionally, a randomized open-label, active-controlled clinical trial found that patients receiving green tea extract in combination with diclofenac experienced significant reductions in pain and notable improvements in physical function [147]. However, other studies did not show significant differences in pain and stiffness sub-scores, suggesting that green tea’s impact on joint stiffness may be limited over short periods [147].

Moreover, green tea was found to be safe, with few reported adverse effects [130], underscoring the potential of green tea as a complementary therapy for OA, and highlighting the need for further long-term studies to confirm its efficacy and establish optimal dosing protocols.

### 3.6. Harpagophytum procumbens

*Harpagophytum procumbens* (HP), commonly known as devil’s claw, is a medicinal plant native to the Kalahari Desert and the steppes of Namibia, Botswana, and South Africa. The complex extract from HP, or devil’s claw, contains phenolic acids and glycosides, triterpenes, phytosterols, iridoid glucosides like harpagoside, and various flavonoids [148]. These extracts are frequently studied for their effectiveness in reducing inflammation, pain, and other symptoms associated with osteoarthritis.

#### 3.6.1. In Vitro

HP exhibits notable anti-inflammatory properties in chondrocytes [149]. The active component harpagoside has been shown to suppress the expression of IL-6 and MMP13 in primary human osteoarthritis chondrocytes [150]. The same group previously suggested that the anti-inflammatory activity of harpagoside is due to its inhibition of the NF-κB pathway [151]. However, Mariano and colleagues argued that the chondroprotective effects observed in in vitro models are due to the combined action of all the individual components in the HP extract and not only to harpagoside [152], and suggest that whole-plant extracts have a better therapeutic effect than those prepared from isolated parts [149].

In human synoviocytes, HP root extract demonstrates significant anti-inflammatory effects by reducing the production of inflammatory mediators such as TNF-α and IL-1β [152].

HP also influences subchondral bone. Studies suggest that it can modulate bone remodeling processes by affecting the activity of osteoblasts and osteoclasts. For instance, the extract has been shown to downregulate RANKL at both mRNA and protein levels, and it regulates the formation of osteophytes, which are bone spurs that develop in response to joint damage in mice OA model. This regulation is mediated through pathways involving TGF-β and BMP-2, which are critical factors in bone formation and resorption [153].

#### 3.6.2. In Vivo

The effect of HE extracts has been very few investigated. Only a recent study from Xu and collaborators demonstrated that harpagide (HPG) prevents articular cartilage degeneration in a rat OA model [154]. OA rats treated with a daily administration of 20 mg/kg of HPG for 8 weeks showed notable improvement in the structure of the articular cartilage compared to untreated rats. Histological evaluations revealed that the HPG-treated group had reduced cartilage degradation. Biochemical analyses showed also that HPG reduced levels of pro-inflammatory proteins such as IL-6 and MMP-13 [154].

#### 3.6.3. Clinical Trials

The clinical effects of HP have been studied to some extent. In a double-blind, randomized, multicenter clinical study conducted by Chantre et al., involving 122 patients with knee and hip OA, the efficacy and safety of Harpadol^®^ (HP) were compared with diacerein over a 4-month period. The study administered a daily dose of 2.6 g of HP extract. The results demonstrated that both treatments significantly improved pain and functional disability. HP was found to be as effective as diacerein, with fewer patients requiring additional NSAIDs and experiencing fewer gastrointestinal adverse events [155].

Similarly, a randomized-controlled clinical trial conducted by Wegener et al., involving patients with knee OA, evaluated the efficacy of Doloteffin™ (HP) at a daily dose of 2.4 g for 12 weeks. This study found significant improvements in pain, stiffness, and physical function, indicating that HP might be an effective alternative treatment with a favorable safety profile [156].

More recently, Farpour et al., investigated the effect of Teltonal^®^ (HP) on knee osteoarthritis in a randomized controlled clinical trial involving 38 patients. The administration of two tablets of HP procumbens (2 × 480 mg) daily for one month showed significant improvement in pain reduction and functional enhancement. The results indicated that HP was as effective as meloxicam, a routine NSAID, in managing symptoms of knee osteoarthritis over an 8-week period. Additionally, patients reported high levels of satisfaction with minimal side effects, suggesting that HP is a viable and safer alternative for those who cannot tolerate NSAIDs [157].

Together, these studies suggest a beneficial action of *Harpagophytum procumbens* for OA patients. However, as noted by Chrubasik et al., there remains a need for standardized outcome measures and rigorous methodologies to confirm the positive effects of *Harpagophytum procumbens* on pain and function in OA patients [158].

### 3.7. Resveratrol

Resveratrol (3,4′,5-trihydroxy-stilbene, RSV) was first extracted from *Veratrum grandiflorum* by Takaoka in 1939 and is widely found in over 70 plants, including grapes, berries, and peanuts [159]. This natural polyphenolic compound has attracted significant attention for its potential joint health benefits, owing to its anti-inflammatory and antioxidant properties.

#### 3.7.1. In Vitro

Resveratrol demonstrates significant anti-inflammatory, anti-apoptotic, and antioxidative effects in chondrocytes. It counteracts the proinflammatory effects of IL-1β by reducing the expression of PGE2, MMP1, MMP3, and MMP13. Beyond its anti-inflammatory and anticatabolic properties, resveratrol also promotes anabolic activity, as it enhances collagen II expression in human OA chondrocytes stimulated with IL-1β [160]. Additionally, resveratrol has been shown to protect chondrocytes from apoptosis by downregulating the COX-2/NF-κB pathway [161]. It also mitigates IL-1β-induced chondrocyte damage through its antioxidative properties, reducing iNOS and ROS production while enhancing cell viability [162].

Resveratrol also disrupts IL-1β-induced pro-inflammatory paracrine interactions between chondrocytes and macrophages, leading to reduced secretion of pro-inflammatory cytokines such as IL-6 and inhibiting the activation of downstream signaling pathways such as STAT3 in macrophages [163]. This suggests that resveratrol can modulate the inflammatory environment within the synovium, potentially decreasing synovial inflammation and joint degradation [162]. Additionally, the inhibition of the NF-κB pathway by resveratrol in rabbit synoviocytes results in decreased production of reactive oxygen species and pro-inflammatory mediators, further protecting the joint environment from inflammatory damage [163,164].

Resveratrol also exerts beneficial effects on osteoblasts, the primary cells responsible for bone formation and remodeling in subchondral bone. It enhances osteoblast activity by upregulating SIRT1 expression, which is a NAD^+^ dependent deacetylase involved in regulating metabolism and promoting cellular health. This increasing in SIRT1 leads to enhanced phosphorylation of Erk1/2, a key signaling pathway that supports cell survival and proliferation. Additionally, resveratrol reduces thex expression of g RANKL, a molecule that promotes osteoclast differentiation, thereby favoring bone formation. Ultimately, these actions contribute to increased mineralization in osteoblasts derived from OA patients [165]. This indicates that resveratrol not only supports bone formation but also counteracts the abnormal activity observed in OA-affected osteoblasts. Furthermore, using hFOB 1.19 cell line, it has been shown that resveratrol reduces osteoblast apoptosis by upregulating anti-apoptotic genes and downregulating pro-apoptotic genes, thereby enhancing cell survival [166].

**Table 1 ijms-25-11656-t001:** In vitro effects of bioactive compounds on different cell types in osteoarthritis models.

Plant Name	Cell Type	Effect	References
*Curcuma longa*	Chondrocytes	↘ MMP-1, MMP-3 and MMP-13; ↘ TNF-α, IL-1β, and IL-6 (inhibition of NF-kB pathway). ↗ type II collagen and aggrecan.	[55]
	Synoviocytes	↘ pro-inflamatory cytokines, ↘ PGE2, ↘ ROS	[59]
	Bone cells	↘ osteoclastogeneis (RANKL pathway), ↗ BMP, ↗ SMAD	[64]
*Boswellia serrata*	Chondrocytes	↘ of MMP-3 and MMP-13, ↗ type II collagen, ↘ TIMP-1 and TIMP-3	[77]
	Synoviocytes	↘ inflamatory cytokine (inhibition of NF-kB pathway), modulation of COX2, ↘ 5-LOX pathway.	[78]
	Bone cells	↘ osteoclastogeneis (RANKL pathway), ↘ VEGF	[79]
*Zingiber officinale*	Chondrocytes	Down regulation of TNF-α, IL-6, and IL-8, inhibits NO production and reduction of PGE2 and ROS	[101]
	Synoviocytes	Inhibits IL-1β, TNF-α, and COX-2	[106]
	SW1353	↘ of p38 and JNK and ↘ of MMP-13	[99]
	RAW264.7 cells	Inhibition of osteoclast differentiation	[109]
Avocado/Soybean Unsaponifiables	Chondrocytes	Inhibition of IL-1β, IL-6, IL-8, PGE2, COX-2, MMP-13 and iNOS	[119]
	Synoviocytes	inhibition of pro-inflamatory cytokine	[118]
	Bone cells	inhibition MMP-13 activity and nitric oxide synthase (iNOS)	[118]
Tea extracts (Epigallocatechin-3-gallate (EGCG))	Chondrocytes	supress production of IL-1β and TNF-α, inhibition of iNOS, ↘ of MMP-13	[133,134,135]
	Synoviocytes	↘ synovial inflamation	[137]
Harpagoside (from *Harpagophytum procumbens*)	Chondrocytes	↘ IL-6 and MMP-13 expression (inhibition of NF-κB pathway)	[150,151]
	Synoviocytes	↘ TNF-α and IL-1β production (anti-inflammatory effect)	[152]
	Bone cells	↘ RANKL (reduces bone resorption), Regulates osteophyte formation (via TGF-β and BMP-2 pathways)	[153]
Resveratrol	Chondrocytes	↘ PGE2, MMP-1, MMP-3, MMP-13; ↘ COX-2/NF-κB pathway; ↘ iNOS and ROS	[162]
	Synoviocytes	↘ ROS and pro-inflammatory mediators (inhibits NF-κB pathway)	[163,164]
	Bone cells	↑ SIRT1 expression, ↑ Erk1/2 phosphorylation and ↘ RANKL	[166]

#### 3.7.2. In Vivo

Zhao et al. have recently summarized the preclinical studies testing the anti-OA effects of resveratrol [167]. Based on 15 studies, they conclude that resveratrol suppresses joint inflammation, marked by reduced pro-inflammatory markers (IL-1, TNFα, IL6, NO), reduces chondrocyte apoptosis and improves cartilage structure in animal models of osteoarthritis. Wei et al. showed, for instance, that resveratrol (50 mg/kg/3 d for eight weeks) significantly lowered the IL-1β, TNFα or NO levels in the knee synovial fluid of OA rats [168]. Jiang et al. also reported that oral administration of resveratrol alleviates osteoarthritis pathology in C57BL/6J mice model induced by a high-fat diet [169]. Protection of cartilage degradation was confirmed in two different post-traumatic OA models in mice [165,170]. Mechanically, they propose that intra-articular injection of resveratrol significantly prevents the destruction of OA cartilage by activating SIRT1 and thereby suppressing the expression of HIF-2α. HIF-2α is a transcription factor that promotes catabolic processes in chondrocytes by upregulating matrix-degrading enzymes such as matrix metalloproteinases (MMPs) and aggrecanases, along with other catabolic factors [170]. Additionally, resveratrol modulates the OPG/RANKL/RANK signaling pathway in subchondral bone, inhibiting abnormal bone remodeling and reducing osteoclast activity [165]. Beneficial activity of resveratrol in joint health has also been shown in rabbit OA models [171].

**Table 2 ijms-25-11656-t002:** Effects of bioactive compounds in preclinical models of osteoarthritis.

Plant Name	Model	Effect	References
*Curcuma longa*	Rats (MIA)	↘ Pro-inflammatory cytokines (IL-6, IL-1β, TNF-α)	[66]
		↘ Inflammatory cell infiltration	[67]
		↘ Bone resorption and osteophyte formation	[68]
*Boswellia serrata*	Rats (MIA)	↘ MMPs, COX-2, 5-LOX and ↘ iNOS	[77]
		↘ Inflammatory mediators (IL-1, IL-6, TNF-α)	[83]
		Reduces cortical bone erosion	[82]
*Zingiber officinale*	Rats (CIA)	↘ MMPs, IL-1β, TNF-α, and PGE2	[110]
		↘ Synovial hyperplasia and inflammatory cell infiltration	[98]
		Protective effects on subchondral bone	[111]
Avocado/Soybean Unsaponifiables	Rats (MIA)	↘ Catabolic factors	
		↑ Anabolic activity (promotes cartilage regeneration)	[118]
	Canine (ACL)	Reduces synovial inflammation	[121]
	Merinos sheep (meniscectomy model OA)	Enhances articular regeneration	[122]
Epigallocatechin-3-gallate (EGCG)	Rats (DMM)	↘ ECM degradation, cartilage inflammation, and cell senescence	[144]
	Canine (OA model)	Reduced pain, improuvement of joint fuction	[140]
	Mice (DMM)	Reduces inflammation in joint synovium, Prevents chondrocyte apoptosis (via Nrf2 pathway)	[141]
	Pig (OA model)	Inhibits osteoclastogenesis, promotes osteoblast differentiation	[142]
Harpagoside (from *Harpagophytum procumbens*)	Rats (OA model)	Prevents cartilage degradation, reduces IL-6 and MMP-13	[154]
Resveratrol	Rats (OA model)	↘ Pro-inflammatory mediators and reactive oxygen species (inhibits NF-κB pathway), ↑ Collagen II expression.	[168]
		↑ SIRT1 expression, ↑ Erk1/2 phosphorylation, ↘ RANKL	[170]
		Inhibits osteoclast differentiation, promotes osteoblast differentiation	[165]

**Table 3 ijms-25-11656-t003:** Clinical studies on the effects of bioactive compounds in osteoarthritis patients.

Plant Name	Sample/Cohort	Supplementation	Effect	References
*Curcuma longa*	*n* = 101 (randomized, placebo-controlled)	500 mg of standardized curcumin extract twice daily	Significant reduction in knee pain, improved physical function, 37% reduced analgesic use compared to 13% in placebo group	[69]
	Meta-analysis	Bioavailable turmeric extract	As effective as paracetamol for pain relief, more effective in reducing CRP and TNF-α	[71,72,73]
	Meta-analysis, 11 RCTs, 1258 participants	Low (<1000 mg)- and high (≥1000 mg)-dose curcumin	Better pain relief than NSAIDs, similar effects across doses, curcumin recommended as adjunctive treatment for knee OA	[74]
*Boswellia serrata*	Meta-analysis, 7 trials, *n* = 545	100–250 mg of BSE for 4 weeks	Effective in reducing pain, improving joint function, and reducing inflammation in OA patients	[76]
	Meta-analysis, 9 RCTs *n* = 712	Aflapin® (BSE suplementation, 20% AKBA)	Effective in managing OA symptoms	[92]
	*n* = 43 (randomized and double-blind)	333 mg of SLBSP (100 mg of BSE) WokVel™ vs. 333 mg of standardized BSE, three time daily for two months	significant symptomatic relief for knee osteoarthritis, improving pain and function scores as measured by the WOMAC and VAS scales	[96]
*Zingiber officinale*	*n* = 43 (randomized, double-blind, placebo-controlled)	30 mL of G-Rup^®^ syrup (containing 150 mg/mL of ginger extract), administered twice daily for twelve week.	Significant improvements in pain scores and physical function compared to placebo	[114]
	Meta-analysis, 3 RCTs, *n* = 330	Ginger supplementation	Reduces pain and enhances quality of life without significant adverse effects	[113]
	*n* = 120 (double-blind, placebo-controlled clinical trial)	500 mg ginger extract twice daily (3 months)	Significant reduction in pain intensity, improved mobility, and reduced TNF-α and IL-1β serum levels	[115]
Avocado/Soybean Unsaponifiables	Meta-analysis, 4 RCTs, *n* = 664	300 mg ASU, Average trial duration was 6 months (range: 3 to 12 months)	Significant improvement in VAS and Lequesne index in knee OA, not hip OA. Safe treatment with no difference in adverse events compared to placebo	[126]
	Meta-analysis, 5 RCTs, *n* = 1095	300–600 mg/day of ASU, 3 months to 3 years	Beneficial effect in symptomatic knee OA but not hip OA, no significant difference in adverse events vs. placebo	[127]
Tea extracts (Epigallocatechin-3-gallate (EGCG))	*n* = 50 (Randomized, open-label, active-controlled clinical trial)	Green tea extract 1500 mg + diclofenac 100mg/day for 4 weeks	Significant reductions in pain, improved physical function, though limited impact on joint stiffness over short periods	[147]
Harpagoside (from *Harpagophytum procumbens*)	*n* = 122 (double-blind, randomized, multicenter)	2.6 g of HP extract (Harpadol^®^) daily for 4 months	Significant improvement in pain and functional disability, as effective as diacerein, fewer gastrointestinal side effects compared to NSAIDs	[155]
	*n* = 75 (Open-label)	2.4 g of HP extract (Doloteffin™) daily for 12 weeks	Significant improvements in pain, stiffness, and physical function, favorable safety profile	[156]
	*n* = 38 (randomized-controlled trial)	Two tablets of HP procumbens (2 × 480 mg) Teltonal^®^ daily for 1 month	Significant improvement in pain and function, as effective as meloxicam, high patient satisfaction with minimal side effects	[157]
Resveratrol	*n* = 142 (randomized placebo-controlled trial)	40 mg twice daily for 1 week, then 20 mg twice daily for 6 month.	No significant difference in knee pain reduction between RSV and placebo groups at 3 and 6 months	[172]

#### 3.7.3. Clinical

In vitro experiments and animal models suggest that RSV has chondroprotective effects, likely due to its ability to reduce the production of inflammatory factors and apoptotic molecules. However, there are still relatively few studies on RSV in patients with osteoarthritis but all have shown good efficacy according a recent review by Yang and collaborators [159]. Nevertheless, the most recent randomized placebo-controlled trial involving 142 patients with painful knee osteoarthritis (71 receiving RSV and 71 receiving a placebo) found no significant difference in knee pain reduction between the resveratrol and placebo groups at 3 and 6 months [172]. This suggests that further randomized controlled trials are necessary to conclusively determine the benefits of RSV for joint health.

### 3.8. Others Bioactive Compounds

Numerous other bioactive compounds from plants have demonstrated significant anti-inflammatory properties, making them potential candidates for OA treatment. For instance, quercetin, a flavonoid found in many fruits and vegetables, inhibits the production of pro-inflammatory cytokines such as IL-1β and TNF-α, and reduces the expression of matrix metalloproteinases in chondrocytes, thereby protecting cartilage from degradation [148]. Similarly, berberine, an alkaloid from Berberis plants, shows potent anti-inflammatory effects by inhibiting the NF-κB pathway, reducing cytokine production, and preventing cartilage degradation [173]. Naringenin, another flavonoid from citrus fruits, shares these anti-inflammatory and chondroprotective properties, promoting the synthesis of cartilage matrix components and reducing MMP expression [174].

In addition to their anti-inflammatory effects, some compounds also exhibit strong antioxidative properties. Baicalin from *Scutellaria baicalensis* demonstrates significant antioxidative activity. Baicalin reduces the production of reactive oxygen species (ROS) and protects chondrocytes from apoptosis, promoting cartilage repair [175]. Diosgenin, a steroidal saponin from fenugreek and wild yam, also contributes to cartilage health by inhibiting the NF-κB pathway and decreasing oxidative stress [176].

Furthermore, other natural compounds, such as pterostilbene and genistein, exhibit both anti-inflammatory and antioxidative properties. Pterostilbene, found in blueberries and grapes, reduces the production of inflammatory mediators like IL-6 and TNF-α, and improves joint function [177]. Genistein, an isoflavone from soybeans, enhances the synthesis of cartilage matrix components and reduces pro-inflammatory cytokine production [178]. These dual actions make them promising candidates for comprehensive OA management.

Saffron (*Crocus sativus*) and *Eugenia caryophyllata* (clove) are notable for their multifaceted effects on OA. Saffron’s bioactive compounds, such as safranal, exhibit anti-inflammatory, antioxidant, and anti-apoptotic properties, alleviating pain and improving joint function [179]. Clove, containing eugenol, inhibits pro-inflammatory cytokines and MMPs, reduces oxidative stress, and protects joint tissues from damage [180].

Bromelain, a proteolytic enzyme complex from pineapples, offers unique anti-inflammatory and analgesic properties. It inhibits pro-inflammatory mediators such as prostaglandins and cytokines, reduces joint swelling and pain, and improves joint function by modulating inflammatory pathways and promoting tissue repair [181].

Collectively, these less commonly used bioactive compounds present promising therapeutic effects for OA through their anti-inflammatory, antioxidative, and chondroprotective actions. Quercetin, berberine, and naringenin focus on reducing inflammation, while baicalin primarily offers antioxidative benefits. Compounds like pterostilbene and genistein provide both anti-inflammatory and antioxidative effects, enhancing overall joint health. Saffron and clove contribute through their multifaceted actions, and bromelain stands out for its unique proteolytic activity. Further research is needed to translate these findings into clinical practice and to determine the most effective

## 4. Discussion

This review highlights the promising potential of several natural bioactive compounds in mitigating osteoarthritis. Compounds such as curcumin, boswellic acids, eugenol, resveratrol, and others mentioned in this review have shown effectiveness in alleviating inflammation, oxidative stress, and cartilage degradation through various mechanisms. They also exhibit additional effects in synoviocytes and subchondral bone remodeling, making them potential options for managing OA symptoms and slowing disease progression. However, despite these promising outcomes, several challenges remain before these compounds can be fully integrated into clinical practice.

### 4.1. Standardization of Plant Extracts and Potential Contaminations

One major challenge in compiling data from various studies on plant extracts is the lack of standardization in the extracts used. The origins of the plants may vary, and the extraction methods also differ between studies, resulting in inconsistencies in the active compounds present in the extracts, which differ from one study to another. This variability makes it difficult to reproduce certain findings and compare results across studies. For example, our review highlights variability in the efficacy and mechanisms of action of *Boswellia serrata* gum resin extracts, which can be attributed to differences in plant origin and extraction processes [83,182,183]. Furthermore, the inherent variability in the composition of natural extracts—due to differences in raw material origine and extraction methodologies-necessitates rigorous verification of the safety and efficacy of plant extracts before their application in humans [83,184]. Additionally, the presence of contaminants can lead to adverse effects, undermining the overall therapeutic value of these extracts [185]. Therefore, strict quality control and precise measurement of active molecules concentrations are essential to ensure the safety and reproducible efficacy of these products.

### 4.2. Cytotoxicity and Safety of Bioactive Compounds

An often underestimated issue with the use of plant extracts is their potential toxicity and side effects. Natural products are frequently considered safe, but at high doses, these substances may cause various side effects or interact with other medications or treatments taken by OA patients. Therefore, it is crucial to exercise caution before and during the use of plant extracts as supplementary medications.

For example, curcumin is usually considered safe because its side effects are not as severe as those of NSAIDs; however, some people might experience nausea or diarrhea after ingesting curcumin [74]. In addition, at high concentrations, curcumin can induce cytotoxicity in chondrocytes [60]. This can compromise cartilage cell viability and exacerbate cartilage degradation if dosages are not properly managed [186]. Similarly, *B. serrata* potentially causes syndrome of inappropriate antidiuretic hormone secretion when it is taken at high doses [187]. In addition, Boswellic acids from *Boswellia serrata*, particularly AKBA, have been shown to cause apoptosis and mitochondrial dysfunction in synovial cells and chondrocytes at high doses [78]. Ginger compounds also present a mixed safety profile. While they exhibit anti-inflammatory and analgesic effects, studies show that high concentrations can induce oxidative stress and cellular damage, particularly in synovial cells [188]. This suggests that while ginger compounds may alleviate OA symptoms, their cytotoxicity at elevated levels warrants further investigation. Regarding to avocado/soybean unsaponifiables, they have generally been associated with low toxicity; however, some adverse effects has been reported which might due to issues in the content or purity of ASU supplements on the market [117]. Similarly, EGCG, known for its potent antioxidant properties, demonstrates a biphasic dose-response relationship. While low doses offer protective benefits against oxidative stress, higher doses may induce cytotoxic effects [189]. Lastly, harpagoside and resveratrol are generally regarded as safe, though some individuals may experience mild gastrointestinal discomfort, especially at higher doses [190,191]. Thus, patients should not self-medicate.

### 4.3. Bioavailability and Pharmacokinetics

Besides the extracted product’s quality, their bioavailability throughout the gastrointestinal system and next in the joint is crucial for their efficiency. Numerous factors affect the bioavailability of compounds. First, many bioactive compounds are poorly absorbed in the gastrointestinal tract due to their large molecular size, poor solubility in water, and instability in the acidic environment of the stomach. For example, curcumin, the active component of turmeric, has low aqueous solubility and a large molecular weight, which limits its absorption across the intestinal epithelium [192]. Next, once absorbed, bioactive compounds often undergo rapid metabolism in the liver and intestines through enzymatic processes such as glucuronidation and sulfation. This rapid biotransformation converts the compounds into more water-soluble metabolites that are quickly excreted from the body. In the case of curcumin, over 90% of the ingested compound is metabolized and excreted within a few hours, leaving only a small fraction available for systemic effects [193]. Additionally, gingerols and their analogues have a low solubility due to their structure, which impacts their bioavailability [97]. A pharmacokinetic analysis revealed a 1–3 h half-life of ginger extract and its metabolites after oral administration [194]. Additionally, in animal studies, 6-gingerol administered intravenously was again more rapidly removed, with a terminal half-life inferior to 10 min [195].

The short half-life of plant compounds in the body necessitates frequent dosing or large doses, which can lead to gastrointestinal discomfort or other side effects, further limiting their clinical utility [196]. To address this limitation, several innovative strategies have been developed. One such approach involves the use of nanotechnology-based formulations, including nanoparticles and nanoemulsions. These nanoformulations enhance the delivery and absorption of bioactive compounds by increasing the surface area for absorption, protecting them from degradation in the digestive tract, and facilitating their transport across biological membranes [197,198]. For example, curcumin-loaded nanoparticles have been shown to significantly improve oral bioavailability, prolong its half-life, and increase its therapeutic efficacy in animal models [199]. Another promising strategy is liposomal encapsulation, which uses spherical vesicles with a phospholipid bilayer to encapsulate bioactive compounds. This method protects the compounds from enzymatic degradation and improves their solubility and stability; liposomal formulations of curcumin, in particular, have demonstrated increased absorption and prolonged circulation time in the bloodstream, thereby enhancing their therapeutic potential [200].

Additionally, co-administration with bioenhancers—substances that improve the absorption and bioavailability of other compounds—has shown effectiveness in overcoming these limitations. Natural bioenhancers such as piperine from black pepper and quercetin have been widely used to enhance the bioavailability of curcumin by inhibiting enzymes that rapidly metabolize these compounds, allowing for prolonged circulation in the bloodstream [201]. Other natural bioenhancers, like resveratrol and epigallocatechin gallate (EGCG) from green tea, have also been used to improve the pharmacokinetic profiles of various bioactive compounds by modulating drug-metabolizing enzymes and transporters [198]. Moreover, forming complexes with phospholipids is another technique to enhance bioavailability [202]. This approach, known as phytosome technology, increases the lipophilicity of bioactive compounds, facilitating their absorption across cell membranes [203]. Phytosome technology has been successfully applied to compounds like curcumin and boswellic acids, resulting in improved bioavailability and therapeutic effects in clinical studies. Nanoformulations, such as nanoemulsions or liposomal encapsulation, have also been utilized for curcumin and boswellic acids, protecting them from degradation and enhancing their stability and absorption [204]. Together, these strategies represent significant advancements in maximizing the clinical potential of bioactive compounds through improved delivery and stability [205].

### 4.4. Limitations in Design of Preclinical and Clinical Studies

The majority of studies investigating the effect of plant extracts in osteoarthritis have limitations.

First, preclinical studies investigating the effect of plant extracts in OA have often been performed in one unique experimental murine OA model, where the disease was induced by MIA injection in joints of young male animals, which does not model the origin of the pathology nor the target population. Given the heterogeneity of profiles in human OA, one animal model is not sufficient to study all features of this disorder [206]. In other words, in order to determine whether a specific therapeutic candidate can be widely applicable for all OA patients, evaluating its efficacy in different models of OA is important as different models represent different phenotypes of the disease. In addition, MIA acts by a non-physiological mechanism: it inhibits the activity of glyceraldehyde-3 phosphate dehydrogenase in chondrocytes, resulting in the disruption of glycolysis, which leads to the hydration of the extracellular matrix, reduction of the synthesis of proteoglycans, chondrocyte death, and eventually cartilage degeneration and subchondral bone damage [207,208,209,210]. Additionally, many animal studies were performed in male animals, limiting the exploration of sex-specific differences in OA pathophysiology and treatment response [211]. In addition, given that elderly women are disproportionately affected by OA, studies should include female to better OA population [212].

Similarly, most current randomized controlled trials (RCTs) have certain limitations, often including missing data on pain, stiffness, and functional indices, as well as small sample sizes. The overall quality of these RCTs tends thus to be moderate to low. In addition, the lack of meta-analyses for some extracts further complicates drawing definitive conclusions. As a result, the findings from clinical studies should always be interpreted with caution.

## 5. Conclusions

In conclusion, natural bioactive compounds show great potential as a multi-targeted approach for managing osteoarthritis (OA). However, there remains a notable lack of robust, objective evidence demonstrating the efficacy and safety of plant extracts in humans for treating or relieving OA. The limited number of randomized clinical trials makes it difficult to draw definitive conclusions regarding the true benefits of these treatments. Nevertheless, as highlighted in this review, several indications point to the potential value of such supplements, particularly in reducing the reliance on anti-inflammatory drugs and delaying the need for joint replacement surgeries.

It is essential that patients adhere to their doctors’ guidance, seeking professional advice before taking these supplements to avoid possible side effects or drug interactions. Additionally, it is crucial that these supplements are not used in isolation but rather in conjunction with non-pharmacological treatments, such as regular physical activity, to maximize their potential benefits.

## 6. Future Directions

As discussed, numerous studies are still needed to identify the most effective plant extracts for alleviating osteoarthritis. A crucial aspect will be conducting rigorous research using well-characterized extracts. Articles should include information about the provenance of the plants used, the extraction methods employed, and the concentration of active substances. This will facilitate better comparison of results and help identify optimal formulations that maximize therapeutic effects while minimizing side effects. Additionally, long-term studies are necessary to gain a clearer understanding of the safety and potential benefits of using plant extracts in OA management.

Current research also aims to combine different plant extracts to enhance their effects. For instance, a recent study has demonstrated the potential complementary and synergistic effects of curcumin and boswellic acids in managing osteoarthritis [213]. Similarly, curcumin can be combined with bromelain and *harpagophytum* to reduce inflammation in osteoarthritic synovial cells [59].

Finally, it is essential not to overlook the role of gender and sex determinants in osteoarthritis. Historically, most animal studies have been conducted using male subjects, limiting the exploration of sex differences in the joint and skeletal effects of plant extracts. Addressing this research gap is critical, particularly given that elderly women are disproportionately affected by OA [214].

## Data Availability

Not applicable.

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
