# Peer review of "Bioactive Compounds in Osteoarthritis: Molecular Mechanisms and Therapeutic Roles"

_ijms, 2024, doi:10.3390/ijms252111656_

Round 1

Reviewer 1 Report

Comments and Suggestions for Authors

I have completed the revision of the manuscript entitled "Bioactive Compounds in Osteoarthritis: Molecular Mechanisms 2 and Therapeutic Roles". The manuscript's structure (introduction, development, discussion, and conclusion) is logical. The introduction provides sufficient background information. According to the commentary, the chapter "Physiopathology of Osteoarthritis" should be completed.

My comments:

1. For some preparations in the descriptions of in vitro studies it is not indicated on which cells the studies were conducted (human, animal?))) - should be supplemented (especially curcumin).

2. Line 306 sentence begins “Numerous clinical trials .......” However, the authors cite only two papers. It would be better to write “several” ….

3. In the descriptions of the mechanisms of action of bioactive compounds, there are concepts, i.e. JNK, SIRT1 HIF-2α, Erk1/2, which were in no way discussed in Chapter 2 - the mechanisms should be supplemented and clarified 

4. Moreover, once “SIRT1” is written in capital letters once in small letters (Sirt1) should be systematized.

Author Response

Thank you very much for taking the time to review this manuscript. Please find the detailed responses below and the corresponding revisions in track changes in the re-submitted files.

  1. For some preparations in the descriptions of in vitro studies it is not indicated on which cells the studies were conducted (human, animal?))) - should be supplemented (especially curcumin).

Answer: We have updated the text to specify the type of cells used in the in vitro studies (human or animal), as suggested. Thank you for the valuable feedback.

  1. Line 306 sentence begins “Numerous clinical trials .......” However, the authors cite only two papers. It would be better to write “several” ….

Answer: We have revised the text to replace "numerous" with "several" as suggested. Thank you for the helpful suggestion.

  1. In the descriptions of the mechanisms of action of bioactive compounds, there are concepts, i.e. JNK, SIRT1 HIF-2α, Erk1/2, which were in no way discussed in Chapter 2 - the mechanisms should be supplemented and clarified

Answer: Thank you for highlighting this point. We agree with the suggestion and have updated the manuscript to include explanations and additional information on each of these proteins (JNK, SIRT1, HIF-2α, ERK1/2) to clarify their roles in the mechanisms of action of bioactive compounds.

  1. Moreover, once “SIRT1” is written in capital letters once in small letters (Sirt1) should be systematized.

Answer: The text has been updated to ensure consistency, with "SIRT1" now always written in capital letters. Thank you for your attention to detail.

Reviewer 2 Report

Comments and Suggestions for Authors

This review summarizes the beneficial effects of various natural compounds in OA and discusses the challenges of incorporating these compounds into clinical practice. The review is well-written and well-structured, making it a valuable resource for readers in the field.

Minor Suggestions:

1) Are Boswellia serrata extracts (BSE) and gum-resin extract of Boswellia serrata (BSRE) referring to the same compounds? If they are the same, it is better to use a consistent abbreviation. Otherwise, please add a brief sentence to clarify the differences.

2) Table 3: There is a typo in the title; "Simple" should be corrected to "Sample" or "Cohort."

Author Response

Thank you very much for taking the time to review this manuscript. Please find the detailed responses below and the corresponding revisions in track changes in the re-submitted files.

1) Are Boswellia serrata extracts (BSE) and gum-resin extract of Boswellia serrata (BSRE) referring to the same compounds? If they are the same, it is better to use a consistent abbreviation. Otherwise, please add a brief sentence to clarify the differences.

Answer: Thank you for bringing this to our attention. We have added a sentence to clarify this point: “Boswellia serrata is frequently available in extract form, which is a concentrated and standardized version of the resin with elevated levels of boswellic acids, especially AKBA, to optimize its anti-inflammatory effects. These extracts are more readily absorbed by the body, making them especially effective for therapeutic use in supplement form.

2) Table 3: There is a typo in the title; "Simple" should be corrected to "Sample" or "Cohort."

Answer: Thank you for the remark. We have corrected the typo, replacing "Simple" with "Sample / Cohort " in the title of Table 3.